# Flare differentially rotates sunspot on Sun's surface

Chang Liu[1,2,3], Yan Xu[1,2,3], Wenda Cao[2,3], Na Deng[1,2,3], Jeongwoo Lee[1,4], Hugh S. Hudson[5,6], Dale E. Gary[3], Jiasheng Wang[1,2,3], Ju Jing[1,2,3] & Haimin Wang[1,2,3]

Sunspots are concentrations of magnetic field visible on the solar surface (photosphere). It was considered implausible that solar flares, as resulted from magnetic reconnection in the tenuous corona, would cause a direct perturbation of the dense photosphere involving bulk motion. Here we report the sudden flare-induced rotation of a sunspot using the unprecedented spatiotemporal resolution of the 1.6 m New Solar Telescope, supplemented by magnetic data from the Solar Dynamics Observatory. It is clearly observed that the rotation is non-uniform over the sunspot: as the flare ribbon sweeps across, its different portions accelerate (up to $\sim 50^\circ\,h^{-1}$) at different times corresponding to peaks of flare hard X-ray emission. The rotation may be driven by the surface Lorentz-force change due to the back reaction of coronal magnetic restructuring and is accompanied by a downward Poynting flux. These results have direct consequences for our understanding of energy and momentum transportation in the flare-related phenomena.

[1] Space Weather Research Laboratory, New Jersey Institute of Technology, University Heights, Newark, New Jersey 07102-1982, USA. [2] Big Bear Solar Observatory, New Jersey Institute of Technology, 40386 North Shore Lane, Big Bear City, California 92314-9672, USA. [3] Center for Solar-Terrestrial Research, New Jersey Institute of Technology, University Heights, Newark, New Jersey 07102-1982, USA. [4] Astronomy Program, Department of Physics and Astronomy, Seoul National University, Seoul 151-747, Korea. [5] School of Physics and Astronomy, University of Glasgow, Glasgow G12 8QQ, UK. [6] Space Sciences Laboratory, University of California, Berkeley, California 94720-5071, USA. Correspondence and requests for materials should be addressed to C.L. (email: chang.liu@njit.edu) or to H.W. (email: haimin.wang@njit.edu).

Sunspots on the solar surface are the most visible manifestation of solar magnetic field[1,2], which has a direct and critical influence on space weather. Line-tied to the dense ($\sim 10^{-7}$ g cm$^{-3}$) photosphere with high plasma beta (ratio of gas to magnetic pressure, $\beta > 1$; $\beta \approx 1$ in sunspots), magnetic fields of sunspots and the induced active regions (ARs) extend into the tenuous ($\sim 10^{-15}$ g cm$^{-3}$) low-beta ($\beta \ll 1$) corona. Thus, the long-term (in days) evolution of photospheric magnetic field, as driven by surface flows and new flux emergence, plays a key role in shaping coronal field structure and, importantly, building up free energy in the corona that powers solar flares via magnetic reconnection[3,4]. For example, the gradual rotational motion of sunspots (generally up to a few degrees per hour) can, in principle, braid and twist the field, leading to an increase of helicity and energy in the corona[5–10]. Sunspots frequently exhibit rotation and this has been linked in the past to the storage of free magnetic energy associated with currents flowing through the corona[11–13].

Once triggered, solar flares give rise to a variety of emission signatures. It is generally accepted that accelerated particles can stream down from the magnetic reconnection site in the corona to the low atmosphere along newly formed magnetic loops, producing chromospheric H-alpha and hard X-ray (HXR) emissions[14]. The former usually appears in eruptive flares as two separating ribbons straddling the magnetic polarity inversion line[15]; the latter is thought to be due to thick-target bremsstrahlung of high-energy particles[16], both reflecting the reconnection process. Subsequently, the heated plasma evaporates to fill flare loops, emitting soft X-rays (SXRs) and other wavelength emissions as it cools. As magnetic flux tubes in the corona are anchored in the dense photosphere, the possibility of a non-particle-related, impulsive (in tens of minutes) and permanent photospheric structure change has been ignored in almost all models of flares and the often associated coronal mass ejections (CMEs), which primarily focus on the coronal field restructuring. Recently, a theory based on momentum conservation predicts that as a back reaction on the solar surface and interior, the photospheric magnetic field would become more horizontal (that is, inclined to the surface) near flaring magnetic polarity inversion lines after flares/CMEs[17,18]. This prediction has been confirmed in multiple observations (for example, see refs 19–22). As the plasma beta within sunspot umbrae and inner penumbrae could be lower than unity[2,23], the Lorentz-force change at and below the photosphere, as quantified by the above back reaction theory, may drive bulk plasma motions in sunspots; however, related supporting observations are extremely rare[24,25]. There is only one study reporting the rotation of a sunspot along with a flare[25], but a definite conclusion on its relationship with the flare emission was hampered by insufficient image resolution.

To advance our understanding of the response of the photosphere to the flare-associated coronal restructuring, here we study the 22 June 2015 M6.5 flare (SOL2015-06-22T18:23) using TiO broadband (a proxy for the continuum photosphere near 7,057 Å) and H-alpha red-wing ($+1$ Å) images with the highest resolution ($\sim 60$ km) ever achieved and rapid cadence (15 and 28 s, respectively). These data are obtained from the recently commissioned 1.6 m New Solar Telescope (NST)[26–29] at Big Bear Solar Observatory (BBSO), which is equipped with a high-order adaptive optics system (see Methods). The high spatiotemporal-resolution imaging capability of NST offers an unprecedented opportunity to investigate the low-atmosphere dynamics in detail. Also used are time profiles of flare HXR and SXR fluxes from the Fermi Gamma-Ray Burst Monitor[30] and the Geostationary Operational Environmental Satellite (GOES)-15, respectively, and photospheric vector magnetograms from the

Solar Dynamics Observatory's (SDO's) Helioseismic and Magnetic Imager (HMI)[31]. With these multiwavelength observations, we clearly see the sunspot in this flaring AR rotating when the flare ribbon propagates through it; more importantly, different portions of the spot accelerate (up to $\sim 50°$ h$^{-1}$) at different times corresponding to the flare HXR peaks. This fast rotation is distinct from the aforementioned slow sunspot rotation seen in the pre-flare stage. As a comparison, the only other similar study[25] used the SDO/HMI intensity data, of which the spatial (temporal) resolution is about 12 (3) times lower than that of the current BBSO/NST data. Our highest resolution makes it possible to resolve the differential sunspot rotation and uncover its intrinsic relationship with the flare emission. We also analyse the flare-related photospheric vector magnetic field change and find that the observed sunspot rotation may be driven by the Lorentz-force change due to the back reaction of coronal magnetic restructuring. Furthermore, we compute the temporal evolution of the energy (Poynting) and helicity fluxes through the surface, and find that they reverse sign during the flare, suggesting that the energy source for the sudden rotation comes from the corona rather than from below the photosphere. These results have direct consequences for our understanding of energy and momentum transportation in the flare-related phenomena.

## Results

**Event overview.** The 22 June 2015 M6.5 flare occurred in NOAA AR 12371 (8°W, 12°N) and was associated with a halo CME. The flare starts at 17:39 universal time (UT), peaks at 18:23 UT and ends at 18:51 UT in GOES 1.6–12.4 keV SXR flux, and has three (I–III) main peaks in Fermi 25–50 keV HXR flux at 17:52:31, 17:58:37 and 18:12:25 UT, respectively. The flare core region was covered by the field of view of BBSO/NST, showing two separating flare ribbons in H-alpha (see Fig. 1a and also Supplementary Movie 1 of ref. 32). The ribbons in TiO are much weaker but still discernible. In particular, the eastern flare ribbon sweeps through the regions of two sunspot umbrae $f1$ and $f2$ of positive magnetic polarity (Fig. 1a). From the movies constructed using the TiO and H-alpha images (Supplementary Movies 1–3), one can clearly find that $f1$ and $f2$ (especially $f1$) exhibit a sudden rotational motion in the clockwise direction closely associated with the flare. Such observation of a sudden sunspot rotation following a flare, with great details revealed in high resolution, was never achieved. Notably, the TiO data are ideal for tracing the photospheric plasma flow motions, especially in sunspot umbrae. Figure 1b shows the flow patterns in $f1$ and $f2$ right before the flare, derived using the differential affine velocity estimator (DAVE)[33] (see Methods). It portrays fine-scale umbral flows, with a general pattern of inward motion[34,35]. The DAVE results allow us to examine the sunspot rotation in a comprehensive way, as described below.

**Flare-induced sunspot rotation.** We study the dynamics of the sunspot (with an emphasis on $f1$) through two data analysis approaches. We pay special attention to the relationship between the sunspot rotation and the flare emission.

First, we evaluate the rotational motion of the whole sunspot in a simplified solid-body approximation. Considering its shape we fit an ellipse to the $f1$ region determined based on the TiO intensity (for example, see Fig. 1c,d and Methods) and plot the temporal evolution of the angle between the derived major axis of the ellipse (for example, yellow and orange dashed lines in Fig. 1c,d) and the horizontal direction as the blue line in Fig. 2. The result shows that $f1$ begins to rotate clockwise as a whole from $\sim 17:56$ UT (about 3.5 min after the HXR peak I) and the

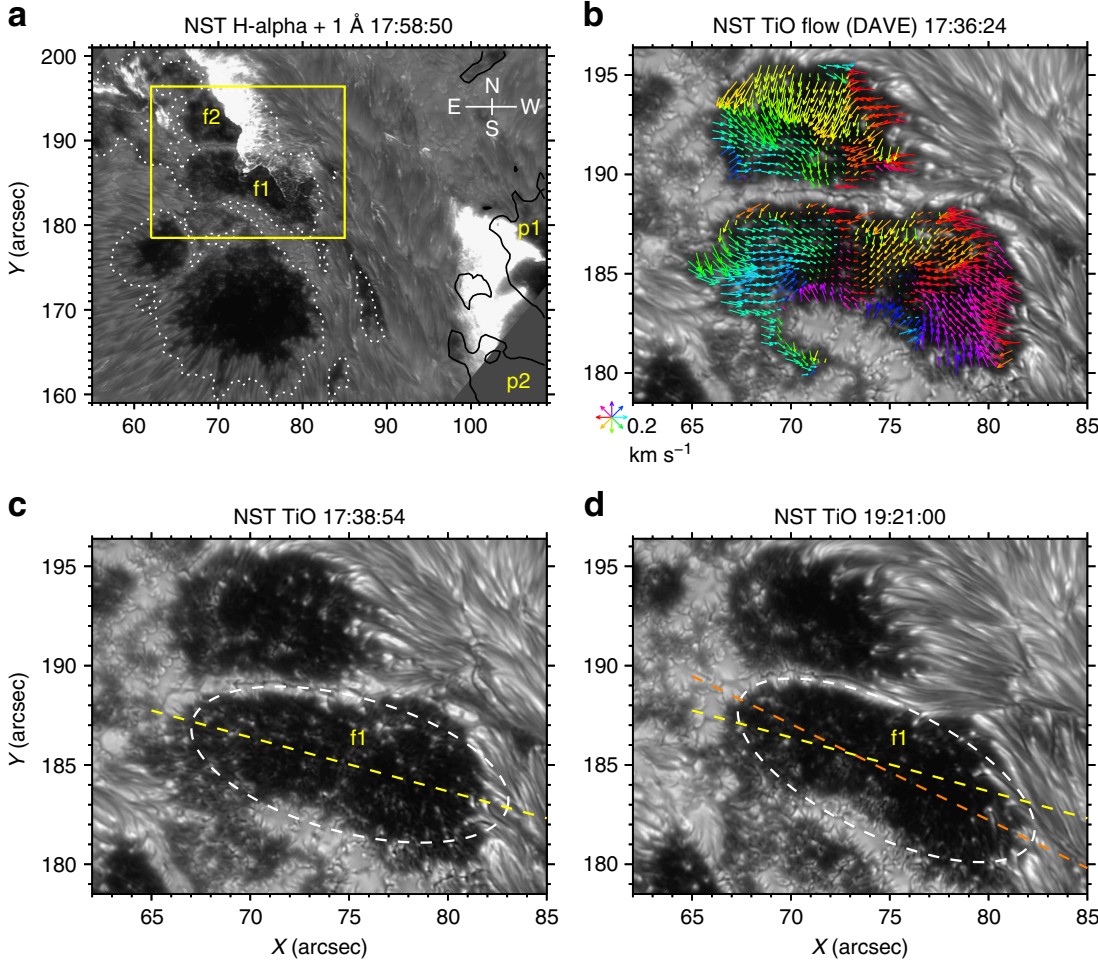

**Figure 1 | Flaring region and sunspot dynamics observed with BBSO/NST.** (**a**) H-alpha +1Å image at the second main HXR peak time showing two separating H-alpha ribbons, with flare-related sunspot umbrae labelled as *f*1, *f*2, *p*1 and *p*2. The white (black) lines contour the 17:58:25 UT vertical magnetic field from SDO/HMI at 1,100 (−1,100) G. The box denotes the field of view of **b**−**d**. (**b**) Pre-flare TiO image superimposed with arrows (colour-coded by direction) representing the flow field in *f*1/*f*2 derived with DAVE (averaged between 17:33:53 and 17:38:54 UT). (**c**) Pre-flare TiO image with the white dashed line representing an ellipse fit to the *f*1 region and the yellow dashed line (also plotted in **d**) the major axis. (**d**) Same as **c** but at a post-flare time, with the major axis drawn in orange.

rotation lasts for about 2 h till ∼20:00 UT, covering a total angular range of ∼13°. Clearly, the present case is distinct from almost all previously studied events, where sunspots undergo a rotation before the flare initiation in SXR. It is also noteworthy that the time profile of the rotation angle can be well approximated by an acceleration function between 17:56 and 18:12:29 UT (around the HXR peak III) followed by a deceleration function (see Fig. 2 and Methods).

Second, a closer examination of the full-resolution movies (Supplementary Movies 2 and 3) unambiguously shows that as the flare ribbon moves across, different portions of the sunspot start rotating at different times (meaning a differential rotation) corresponding to the peaks of HXR emission. To characterize in detail the non-uniform rotation, we resort to the tracking of photospheric plasma flows with DAVE throughout the event (see Fig. 3 and Supplementary Movie 4). Based on the derived velocity vectors, we also compute the flow vorticity (curl of the velocity; calculated by equation (1) in Methods) and examine the spatial and temporal evolution of the negative vorticity (corresponding to a clockwise rotation) in the sunspot region (see Figs 4 and 5, and Supplementary Movie 5). Furthermore, we remap TiO images to a polar coordinate system and trace several distinct features (see Methods and Fig. 6) for a precise

determination of the timing relationship between the sunspot rotation and flare emission. Below, we divide the whole event into three phases and describe the characteristics of sunspot rotation in each phase.

Phase 1 (from HXR peak I at 17:52:31 UT to peak II at 17:58:37 UT): the flare ribbon propagates towards *f*1/*f*2 and just enters into their regions from the west at the time of the HXR peak I (see Fig. 4a). Immediately, the sunspot umbrae underlying the ribbon begin to rotate southwestward. This is clearly exhibited by the space-time slice image (Fig. 6a) from the re-mapped TiO images along the circle C1 (in Fig. 3b), in which the northeastern portion of *f*1 (as represented by features 1–4, which are co-spatial with the flare ribbon at this time; see Fig. 4a) starts rotating right after the HXR peak I, at a mean angular velocity of 50° h⁻¹. Later, as the ribbon proceeds (Fig. 3a) the far western portion of *f*1/*f*2 seemingly forms a clockwise rotational pattern, which can be visualized by the average flow field in this phase (Fig. 3b). The mean angular velocity of *f*1 reaches a maximum of ∼38° h⁻¹ at 17:56:23 UT (Fig. 4b), about 4 min after the HXR peak I (Fig. 5). It is pertinent to point out that the afore-described ellipse fitting under a solid-body assumption shows a significant rotation of *f*1 only after ∼17:56 UT. This highlights the differential nature of this sunspot rotation.

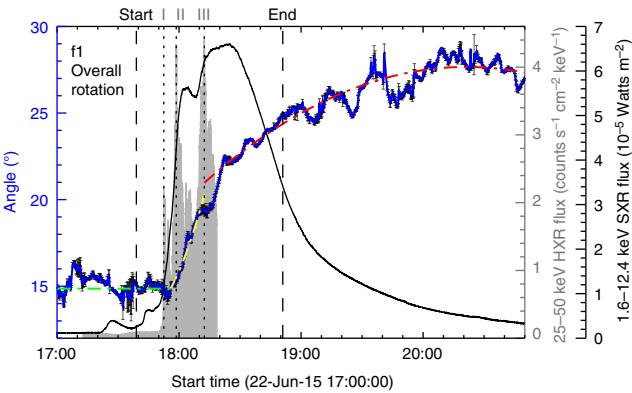

**Figure 2 | Overall sunspot rotation.** Time profiles of SXR flux (black line), HXR flux (gray shaded area; not available during 18:19–19:19 UT) and orientation angle θ of f1 (between the major axis and horizontal direction; blue line) from an ellipse fit (see, for example, Fig. 1c,d). The intensity threshold for delineating the f1 region was varied to evaluate the 1-s.d. error bars of θ. Overplotted is the approximation of θ evolution using a horizontal line between 17:00 and 17:56 UT (green), a second-order polynomial (acceleration) between 17:56 and 18:12:29 UT (yellow), and another second-order polynomial (deceleration) between 18:12:29 and 20:50 UT (red). See Methods for details. The vertical dashed lines mark the start and end times of the flare in GOES 1.6–12.4 keV SXR flux and the dotted lines mark the three main Fermi 25–50 keV HXR peaks I-III.

Phase 2 (from HXR peak II at 17:58:37 UT to about peak III at 18:12:25 UT): the flare ribbon, mainly its northern part, moves a significant distance towards the east, across the main regions of f1/f2 (Figs 3c and 4c–f). As can be seen in Fig. 6, the southern and eastern portions of f1, represented by features 5–7 and 8–10 marked in Figs 3b,d and 4c–e, begin a rotation-like motion immediately following the HXR peak II, at a mean angular velocity of 52° and 30°h$^{-1}$, respectively. It can also be noticed that the northwestern portion of f1 (for example, features 1–4) keeps rotating in this phase. As a result, the entire f1 and f2 display a rotational flow pattern in the clockwise direction (Fig. 3d). The mean angular velocity of f1 has the second maximum of 36° h$^{-1}$ at about 4 min after the HXR peak II and sustains roughly this speed till about 18:08 UT. As for f2, its clockwise rotation keeps accelerating after the HXR peak I, and peaks at 45° h$^{-1}$ about 3.5 min after the HXR peak II (see Fig. 5).

Phase 3 (from about HXR peak III at 18:12:25 UT): the flare ribbon almost moves out of the sunspot region (Fig. 3e). The rotational flows involving both f1 and f2 diminish, as reflected by the observations that the mean vorticity of f1/f2 largely returns to the pre-flare level (Fig. 5), and that drifting features nearly flattens in the re-mapped space-time slice images (Fig. 6). Interestingly, f1 shows overall westward and southwestward flows (Fig. 3f), and it continues to rotate clockwise as a whole (see Fig. 2 and Supplementary Movie 1).

Taken together, the exceptionally high-resolution observations from BBSO/NST make it possible to witness, for the first time, a sudden sunspot differential rotation that exhibits an intrinsic spatiotemporal relationship with the coronal energy release process, manifested as flare ribbon propagation and HXR emission profile. The measured angular velocity of rotation amounts up to ~50° h$^{-1}$, which is much higher than that of the reported pre-flare rotating sunspots. These strongly indicate that the observed sunspot rotation on the photosphere is a result, not a cause, of the flare magnetic reconnection in the corona, which challenges the conventional view of the photosphere-corona coupling.

It is worth noting that similar to the propagating ribbon, the negative vorticity feature also progresses from west to east across the sunspot (see Supplementary Movie 5, vorticity evolution). More exactly, the development of regions of intense negative vorticity follows the flare ribbon motion and concentrates on the portion swept by the ribbon (see Fig. 4). This implies that the sunspot rotation is intimately linked to the flaring process. The features 1–7 in the west start rotating as the flare ribbon sweeps by and ensuing the peaks of the HXR emission (Figs 4a,c and 6). In contrast, features 8–10 in the east begin to move northeastward (with little rotation, that is, low vorticity) at the HXR peak II (Figs 4c and 6), when the ribbon has not spread to their locations. Enhancement of the negative vorticity in these regions occur only when the ribbon arrives ~5 min later (Fig. 4d,e). These two movement stages of the eastern part of f1 are discernible in the time-lapse movie (Supplementary Movie 2). For simplicity, we still describe the earlier motions of features 8–10 as rotations. The umbrae f1/f2 gain maximum angular velocity in a few minutes after the initiation of rotation of sunspot features, consistent with the low Alfvén speed of the photospheric plasma (~10–20 km s$^{-1}$ in sunspot umbrae). Unlike f1, no obvious internal rotations are observed within f2; in fact, together they present a coherent rotation (Fig. 3d), despite of the sunspot light bridge lying between them. This connotes that f1 and f2 may be parts of a unified magnetic structure. As the rotational motion of the whole sunspot shows a deceleration after 18:12:29 UT (Fig. 2), phase 3 could be an after-effect following phase 1 and phase 2 of the rapid rotation directly related to the flare.

**Flare-related magnetic evolution.** As moving H-alpha ribbons are regarded as a mapping of the reconnecting coronal magnetic field onto the low solar atmosphere[14] and HXR emissions could gauge the magnitude of coronal magnetic reconnection[3], the revealed correlation between the sunspot rotation and flare emissions motivates us to explore the changes of magnetic field and related quantities, which can shed light on the mechanism of the flare-induced sunspot rotation. To analyse the photospheric magnetic field and its evolution, we use vector magnetograms from SDO/HMI with 12 min cadence and 1 arcsec spatial resolution (see Methods). We observe that the flare causes apparent changes of the sunspot (especially f1) structure, in terms of intensity and vector magnetic field (see Supplementary Fig. 1). Here we mainly concern ourselves with the Lorentz-force change exerted at and below the surface by coronal magnetic field from above, which is attributed to the restructuring of coronal magnetic field in the back reaction theory[17,18]. There are two HMI measurements made during the main phases of sunspot rotation. At 18:00:44 UT (1.5 min into phase 2), the density map of the horizontal component of the Lorentz-force change $\delta\mathbf{F}_h$ (calculated using equation (2) in Methods) is presented in Fig. 7a. It is remarkable that $\delta\mathbf{F}_h$ forms a swirl in the western portion of f1 and also exhibits a coherent clockwise rotation over regions of f1/f2, resembling a combination of TiO flow patterns of phase 1 and phase 2 (see Fig. 3b,d). As shown in Fig. 7b, the $\delta\mathbf{F}_h$ density map at 18:12:44 UT (beginning of phase 3) changes to an overall rotating structure also similar to the flow pattern of phase 3 (Fig. 3f). Intriguingly, similar to the flow vorticity (Fig. 4) the $\delta\mathbf{F}_h$ distribution seems to evolve with the ribbon motion; however, this aspect needs to be further addressed when higher cadence vector magnetograms become available. In any case, these hint that the torque $T$ produced by $\delta\mathbf{F}_h$ may drive the sunspot rotation, a scenario also suggested by the only other related study[25]. For simplicity, ignoring the differential rotation but assuming a rigid rotation of the elliptical f1 around its centre (cross in Fig. 7), the time profile of $T$ on f1, as plotted in Fig. 8a,

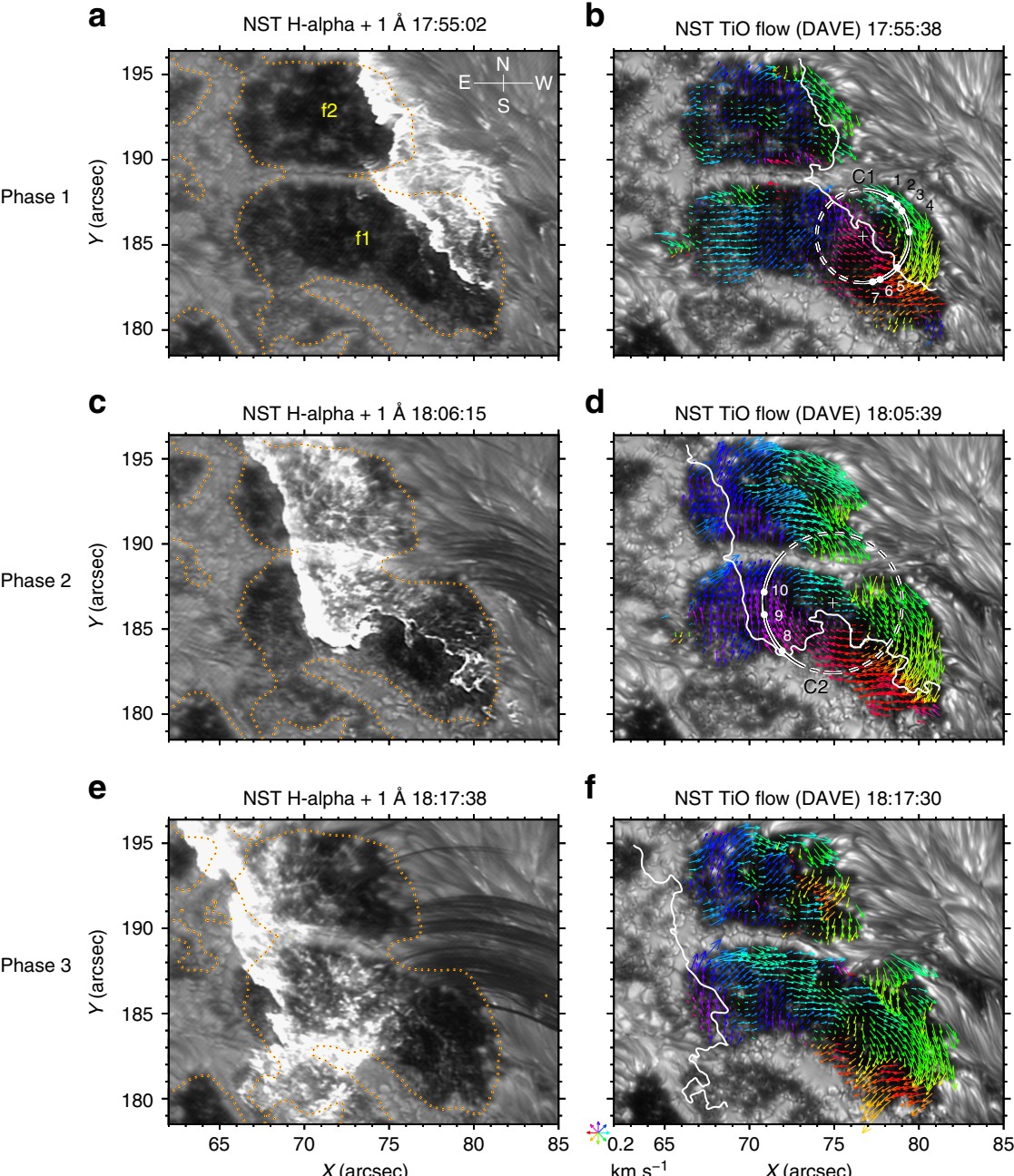

**Figure 3 | Solar flare and induced sunspot rotation.** BBSO/NST H-alpha +1Å images (**a**,**c**,**e**) and the co-temporal TiO images (**b**,**d**,**f**), showing the sunspot rotation in three phases (see text for details and Supplementary Movies 1–4 for animations). SDO/HMI vertical magnetic field is contoured at 1,300 G on H-alpha images. In **b**,**d**,**f**, the superimposed arrows (colour-coded by direction) illustrate DAVE flows in f1/f2 averaged between 17:52:38–17:58:38 UT (phase 1), 17:58:38–18:12:29 UT (phase 2) and 18:12:29–18:22:30 UT (in phase 3), respectively, subtracted by a pre-flare flow field averaged between 17:32:23 and 17:52:23 UT to better show the rotational motion. The overplotted white curves delineate the co-temporal H-alpha flare ribbons. The plus in **b** (**d**) is the origin for the polar re-mapping, with the circle C1 (C2) denoting the constant radius for constructing the space-time slice image presented in Fig. 6a (6b). The angle starts at due South and increases anticlockwise. The beginning angle locations of features 1–10 along C1/C2 as seen in Fig. 6 are marked as solid dots.

shows impulsive $T$ signals closely associated with the rotation of f1. A rough quantitative estimate also indicates that the amount of $T$ on f1 is sufficient compared with that required for the measured rotation (see Methods). The torque rapidly decrease to zero soon after the beginning of phase 3. Thus, the torque evolution is also in line with the observed acceleration followed by deceleration of the overall sunspot rotation (Fig. 2).

With SDO/HMI vector magnetic field data, we further track the photospheric plasma flows using the DAVE for vector magnetograms (DAVE4VM)[36] (see Methods), which can derive not only the horizontal but also the vertical component of flows. These vector photospheric velocity fields permit an accurate assessment of the Poynting flux $\dot{E}$ and helicity flux $\dot{H}$ transported through the photosphere, which are physical quantities intimately associated with rotating sunspots[5–10], thus may help elucidate the essential physics needed to properly interpret our observations. The temporal evolution of $\dot{E}$ and $\dot{H}$ throughout the flare (calculated by equations (3) and (4) in Methods) is drawn in

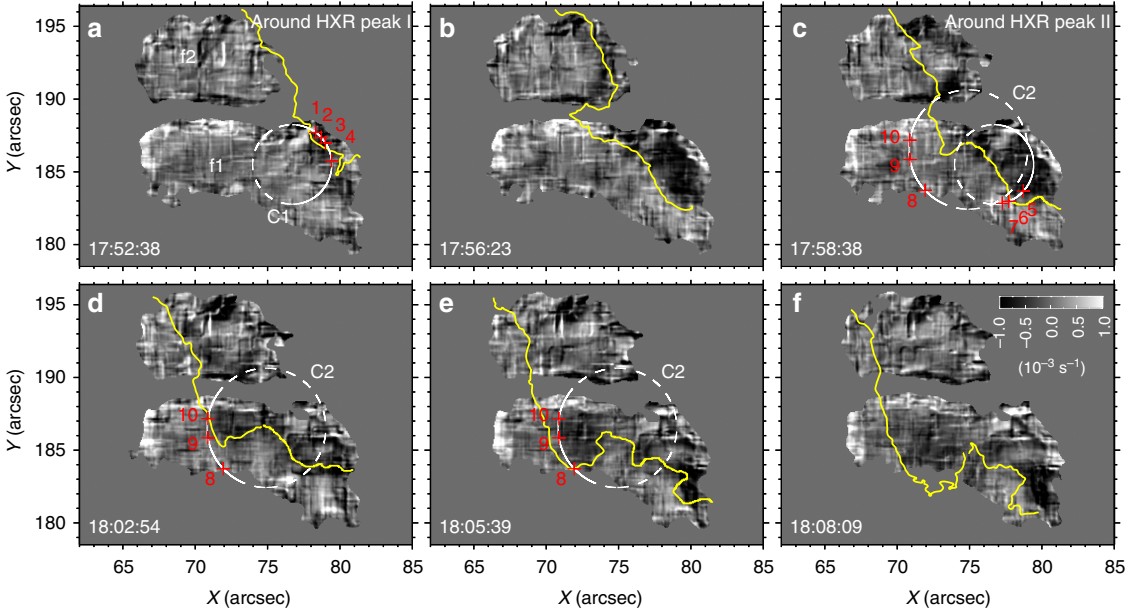

**Figure 4 | Spatial evolution of vorticity.** Time sequence of vorticity maps during phase 1 (**a,b**) and phase 2 (**c–f**) in the regions of umbrae f1 and f2, computed based on BBSO/NST TiO images (see Methods and Supplementary Movie 5 for an animation). The overplotted yellow line denotes the front edge of the co-temporal H-alpha flare ribbon. The circles C1 and C2 and the associated features (crosses 1–10) are the same as those in Fig. 3b,d.

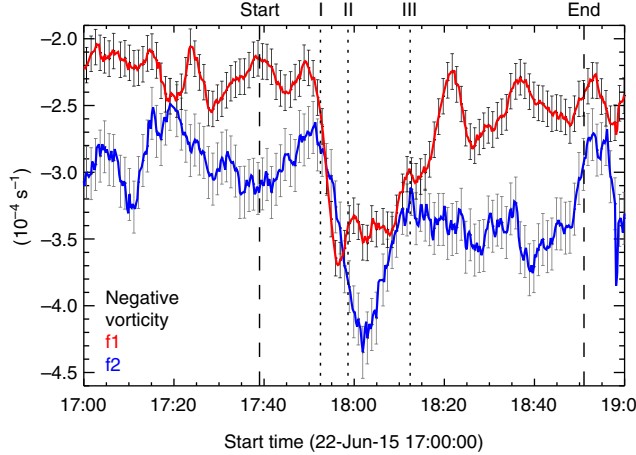

**Figure 5 | Temporal evolution of vorticity.** Mean negative vorticity $\bar{\omega}$ of f1 (red) and f2 (blue). The error bars (plotted every 1 min to better show the results) represent 1 s.d. calculated from the average $\bar{\omega}$ over a pre-flare period (17:00 to 17:39 UT), demonstrating the significant flare-related variations compared to that seen in the long-term evolution. The vertical dashed lines mark the start and end times of the flare in GOES 1.6–12.4 keV SXR flux and the dotted lines mark the three main Fermi 25–50 keV HXR peaks I–III.

Fig. 8b,c. The former is integrated over the regions of f1 and f2, considering the low cadence of HMI data and the fact that f1/f2 could make up a unified magnetic structure (see previous discussion). The latter is integrated over the entire AR. It can be seen that energy and negative helicity are injected upward from below the surface both before and after the flare. The negative sign of helicity conforms with the measured left-handed twist of f1 and f2. However, during the flare time interval, both $\dot{E}$ and $\dot{H}$ reverse sign. In particular, there is a downward Poynting flux during the flare time interval (with a total energy about $1.6 \times 10^{30}$ ergs), which could be the energy source driving the photospheric motion. These point to a physical process associated

with the sunspot rotation (presumably the back reaction of coronal magnetic reconfigurations) that contrasts with that in the non-flaring period.

## Discussion

Our observations demonstrate that sunspots f1/f2 rotate as a response of the flare energy release, and that the rotation is progressive and differential, ensuing the flare emissions. We notice that f1 and f2 are at the footpoints of erupting flux loops, which develop into a halo CME accompanying the present flare. These loops connect to two other sunspots p1 and p2 in negative field regions (Fig. 1a), which vaguely show a similar flare-related clockwise rotation in SDO/HMI data (details, however, are unknown as p1/p2 are out of the field of view of BBSO/NST). This alludes to the possibility that on the large scale, the observed sunspot dynamics may be linked to the properties of a twisted flux tube. With related to sunspot rotation, let us consider theoretically the emergence of a vertical, twisted magnetic flux tube from the interior into the corona[37,38]. During its emergence, rapid expansion and stretching occur to the coronal portion of the tube, where the twist rate of the field ($\alpha = \mathbf{J} \cdot \mathbf{B}/B^2$) decreases rapidly. As a result, along the field lines a gradient of the twist rate gets established, and it drives torsional Alfvén waves that propagate twist from the interior into the corona, until a twist balance is reached on a time scale of a few days. This constitutes an explanation of rotating sunspots in emerging flux regions (for example, see refs 8,39). However, if an eruption suddenly happens that stretches out the coronal field again, the gradient of twist rate and hence the torque on the photosphere would increase, which can consequently cause a sudden increase of the sunspot rotational motion in the same direction as before the eruption, as seen in the only other observation of a flare-related sunspot rotation[25]. Under this scenario, it would be expected that the Poynting flux $\dot{E}$ and also helicity flux $\dot{H}$ (with the same sign as that before the eruption) injected into the atmosphere by the emerging flux tube would also suddenly enhance[40]. However, we observe the exact opposite behaviours of $\dot{E}$ and $\dot{H}$ during this eruptive flare event.

Therefore, we are led to conclude that the driving agent behind and the energy source of the observed sunspot rotation originates

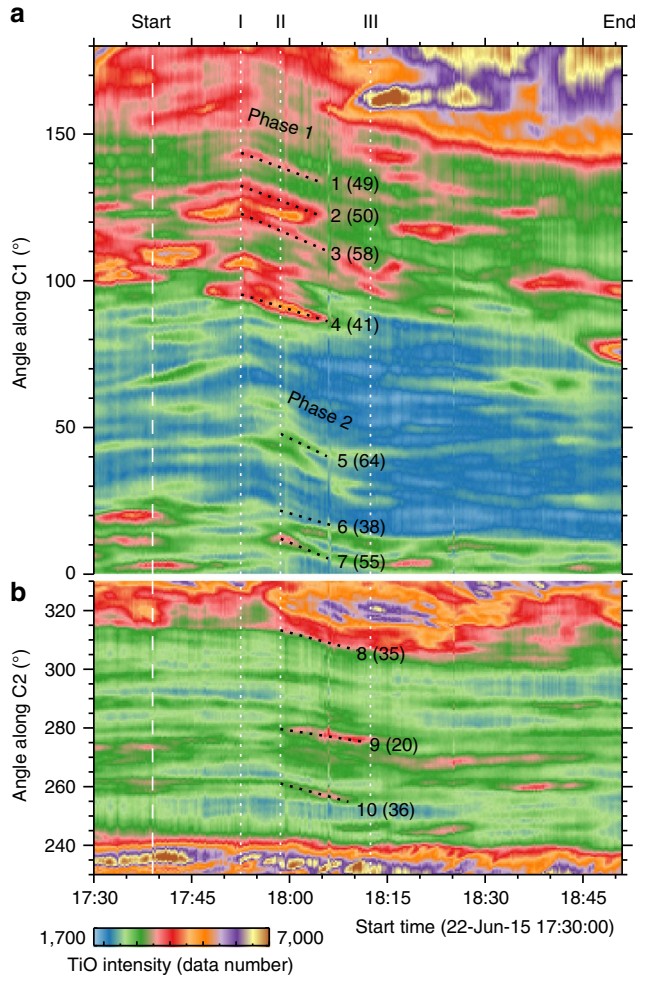

**Figure 6 | Space-time slice image for sunspot rotation.** The results in **a** and **b** are constructed from TiO images re-mapped to a polar coordinate system, at a constant radius of 2.7″ (C1 in Fig. 3b) and 4.1″ (C2 in Fig. 3d), respectively. The shown angular range is 0–180° for C1 and 230–330° for C2, and these ranges are denoted using solid lines when drawing C1/C2 in Figs 3b,d and 4a,c–e. The black dotted lines trace several distinct features 1–10 in *f*1 by a linear approximation. The numbers in bracket are the corresponding angular velocity (in degree per hour) from the linear fit. The initial locations of these features are also indicated in Figs 3 and 4. The vertical dashed lines mark the start and end times of the flare in GOES 1.6–12.4 keV SXR flux and the dotted lines mark the three main Fermi 25–50 keV HXR peaks I–III.

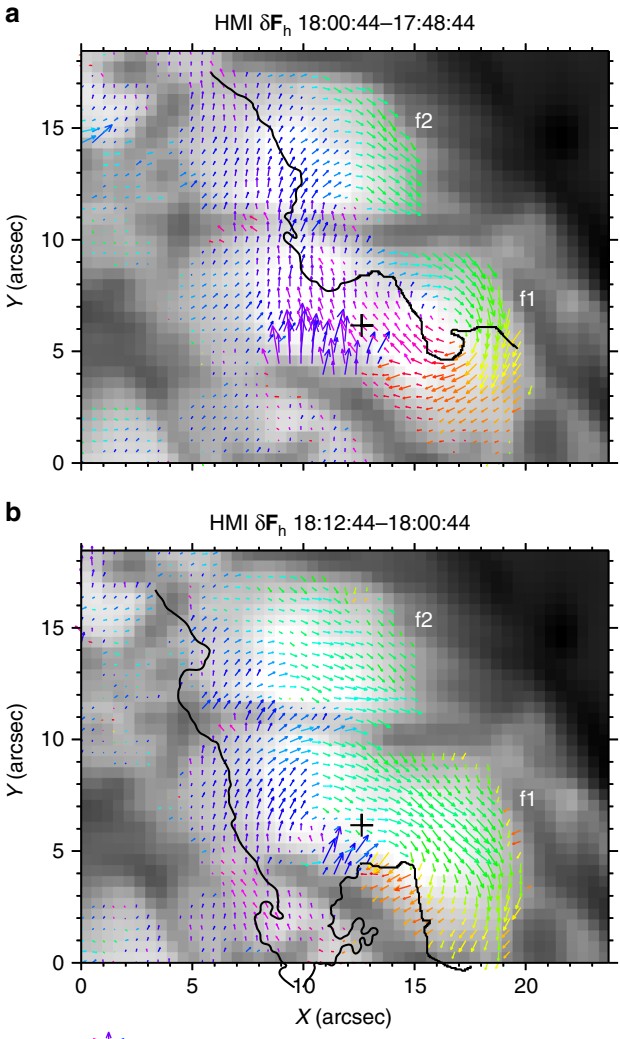

**Figure 7 | Horizontal Lorentz-force change.** SDO/HMI vertical magnetic field, with the white (black) colour representing positive (negative) polarity, superimposed with arrows (colour-coded by direction) displaying the horizontal Lorentz-force change vectors between 17:48:44 and 18:00:44 UT (**a**), and between 18:00:44 and 18:12:44 UT (**b**). The projected and re-mapped HMI data product is used. See Methods for details. Arrows are only shown at locations with vertical field >1,200 G. The cross is the fitted centre of the elliptical *f*1 for the torque calculation shown in Fig. 8a. The black line illustrates the front of the co-temporal H-alpha flare ribbon.

from the corona rather than below the photosphere, most probably associated with the back reaction of the flare-related restructuring of coronal magnetic field. We also postulate that the torque produced by coronal transients might drive the low atmosphere down to a certain depth. Certainly, more observations of the low solar atmosphere in high resolution, together with simulations of photospheric sunspot dynamics[41] and further understanding of the photosphere-corona coupling, are desired to tackle the problem of energy and momentum transportation in the flare-related phenomenon.

## Methods

**Instrumentation and data.** The broadband TiO and H-alpha red-wing images used in the present study, with a spatial resolution of ~61 and 66 km and a cadence of 15 and 28 s, respectively, are obtained with the 1.6 m BBSO/NST, which is currently the largest-aperture ground-based solar telescope. It combines a high-order adaptive optics system using 308 sub-apertures and the post-facto

speckle image reconstruction techniques to achieve diffraction-limited imaging of the solar atmosphere. The H-alpha data are taken by the Visible Imaging Spectrometer, which is a Fabry–Pérot filter-based system that can scan in the wavelength range of 5,500–7,000 Å. For this observation run, five points were scanned around the H-alpha line centre at ±1.0, ±0.6 and 0.0 Å. For data processing, the images were aligned with sub-pixel precision and the intensity was normalized to that of a quiet-Sun area. The TiO and H-alpha images were co-aligned by matching sunspot and plage areas, with an alignment accuracy of about 0.2 Mm. All the images presented in this paper were registered with respect to 22 June 2015 17:38:54 UT.

For the analysis of photospheric magnetic field, we use the observation from HMI on board SDO with 12 min cadence and 1 arcsec spatial resolution. Specifically, for the context study in Figs 1 and 3, and Supplementary Fig. 1, we use the full-disk HMI vector magnetogram data product hmi.B_720s (refs 31,42). For the calculation of Lorentz-force change, tracking of plasma flows with DAVE4VM and computation of Poynting and helicity fluxes, we use the Space-weather HMI Active Region Patches vector magnetogram data product hmi.sharp_cea_720s (ref. 43). The Space-weather HMI Active Region Patches data are re-mapped using Lambert (cylindrical equal area) projection centred on the studied AR.

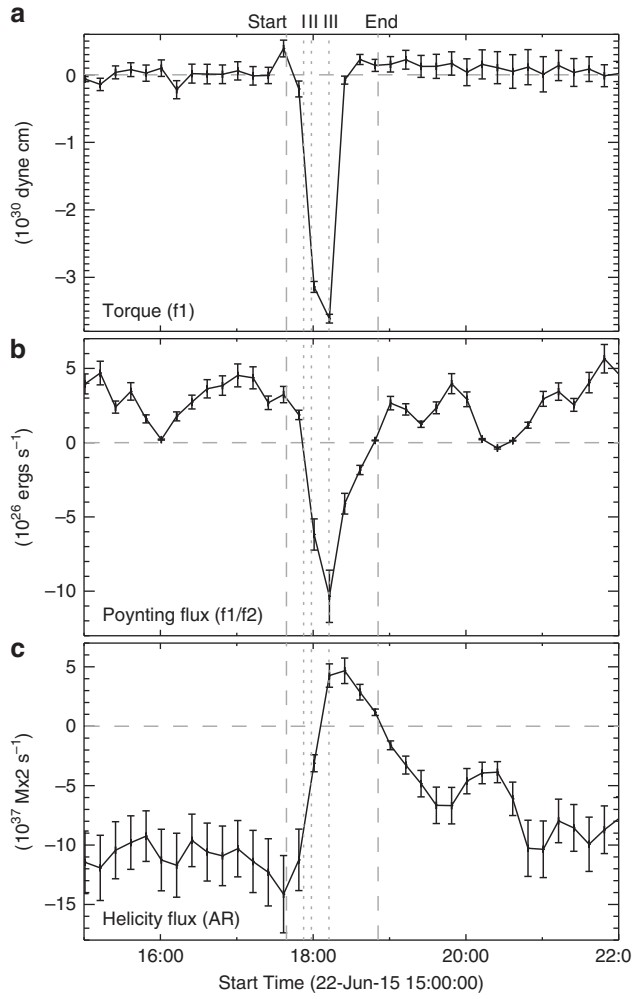

**Figure 8 | Temporal evolution of magnetic properties.** (**a**) Torque on $f1$ resulted from horizontal Lorentz-force change. The error bars represent 1 s.d. calculated from the provided uncertainty of HMI vector field. (**b**) Poynting flux integrated over the rotating sunspots $f1$ and $f2$. (**c**) Magnetic helicity flux integrated over the whole flaring AR. Error bars in **b** and **c** represent an uncertainty of 17% for energy flux and 23% for helicity flux due to noise in the HMI data. See Methods for details. The vertical dashed lines mark the start and end times of the flare in GOES 1.6–12.4 keV SXR flux and the dotted lines mark the three main Fermi 25–50 keV HXR peaks I–III.

**Sunspot rotation analysis.** To evaluate the overall rotation of $f1$, we (1) use the REGION_GROW function in IDL with a pre-set TiO intensity threshold to define the region of $f1$, (2) conduct an ellipse fit to the $f1$ region using the FIT_ELLIPSE function in IDL and (3) vary the intensity threshold from 3,900 to 4,000 data number and perform a total of 11 runs of calculation for error estimation. These threshold values are selected so that the umbra $f1$ can be well delineated throughout the studied time period. The temporal evolution of the angle $\theta$ between the major axis of the fitted ellipse and the horizontal direction, as shown in Fig. 2, is approximated using a least-squares fit to a horizontal line between 17:00 and 17:56 UT, a second-order polynomial $\theta = 14.9 + 3.63 \times 10^{-3} t + 1.84 \times 10^{-6} t^2$ between 17:56 and 18:12:29 UT where $t$ is in units of second from 17:56 UT, and another second-order polynomial $\theta = 21.0 + 1.72 \times 10^{-3} t - 1.11 \times 10^{-7} t^2$ between 18:12:29 and 20:50 UT (the end of this BBSO/NST observation run) where $t$ is in units of second from 18:12:29 UT.

To track the photospheric plasma flows, we employ the DAVE method, which is a well-established, state-of-the-art technique using the advection (adopted here) or continuity equation and a differential feature tracking algorithm for flow detection. In this study, a $2 \times 2$ binning is applied to the TiO data to increase the S/N ratio. The tracking window size is set to 23 pixels, which balances the needs for including enough structure information and a good spatial resolution. We then calculate the vorticity $\omega$ (in units of

$(s^{-1})$) as:

$$\omega = \frac{\partial}{\partial x} v_y - \frac{\partial}{\partial y} v_x, \tag{1}$$

where $v_x$ and $v_y$ are velocity vectors after a 5 min running average of the DAVE flow fields, which is to alleviate the effects of the atmospheric disturbances and photospheric 5 min oscillation contained in the observation. In this definition, vorticity is equal to twice the angular velocity.

Re-mapping of TiO images to a polar coordinate system is carried out with the centre of the rotational flow pattern (plus signs in Fig. 3b,d) as the origin, where the two axes of the re-mapped frames represent the polar angle around and the distance $R$ from the origin. To construct the space-time slice images shown in Fig. 6, we stack one slice per frame, which is averaged for 11 pixels between $R - 0.17''$ and $R + 0.17''$, where $R = 2.7''$ ($4.1''$) for the circle C1 (C2) drawn in Fig. 3b(3d). The size and location of these circles are determined in such a way that the right (left) half of C1 (C2) closely follows the rotational flows in the western (eastern) portion of $f1$ during phase 1 (phase 2).

**Magnetic evolution analysis.** The change of the horizontal Lorentz force exerted at and below the photosphere can be formulated as:

$$\delta\mathbf{F}_h = \frac{1}{4\pi} \int dA \delta(B_r \mathbf{B}_h), \tag{2}$$

where $B_r$ is the photospheric vertical magnetic field and $\mathbf{B}_h$ is the horizontal field vector[17,18]. Assuming that $f1$ has a geometry of rigid elliptical disk rotating about its centre, the torque $T$ resulted from $\delta\mathbf{F}_h$ can produce an angular acceleration $\alpha = T/I = T/\{\frac{1}{4}\rho\pi hab(a^2 + b^2)\}$, where $I$ is the moment of inertia relative to its center, $\rho$ is the photospheric density, $h$ is the depth (a coherent depth of rotation is presumed), and $a$ and $b$ are the length of the semi-major and semi-minor axes of the ellipse that can be derived from the shape fitting. Here we take $\rho \approx (4–11) \times 10^{-7}$ g cm$^{-3}$, $h \approx 270$ km (a density scale height at the photosphere), $a \approx 6.8$ Mm and $b \approx 3.2$ Mm. At 18:00:04 UT in phase 2, the clockwise torque exerted on $f1$ (relative to the centre marked as the cross in Fig. 7) produced by $\delta\mathbf{F}_h$ (relative to a pre-rotation time 17:48:44 UT) amounts to $T \approx 3.1 \times 10^{30}$ dyne cm (Fig. 8a), which can produce an $\alpha$ of $(1.1–3.0) \times 10^{-6}$ rad s$^{-2}$. This is more than sufficient compared to the observed $\alpha \approx 2.3 \times 10^{-7}$ rad s$^{-2}$, when considering that the angular velocity of $f1$ increases $\sim 6.8 \times 10^{-5}$ rad s$^{-1}$ from $\sim$17:51:30 to 17:56:23 UT in phase 1 (Fig. 5). In addition, if considering a total angular distance of $\sim 5°$ till the end of phase 2 (Fig. 2), the work done by the torque (that is, the rotational kinetic energy of $f1$) is roughly $3 \times 10^{29}$ ergs. We caution that our calculation has a large uncertainty due to the assumption of $h$ and ignorance of the differential rotation nature of $f1$.

The DAVE4VM technique based on the magnetic induction equation is employed to track both the horizontal and vertical components of the photospheric plasma flows. For this analysis, we use time series of SDO/HMI data with a window size of 19 pixels, which is selected according to previous studies[44,45].

The vertical component of Poynting flux across the plane $S$ at the photospheric level can be derived as[46]:

$$\left.\frac{dE}{dt}\right|_S = \frac{1}{4\pi} \int_S B_t^2 V_{\perp n} dS - \frac{1}{4\pi} \int_S (\mathbf{B}_t \cdot \mathbf{V}_{\perp t}) B_n dS, \tag{3}$$

where $B_t$ and $B_n$ are the tangential (horizontal) and normal (vertical) magnetic fields, and $V_{\perp t}$ and $V_{\perp n}$ are the tangential and normal components of velocity $V_\perp$ (the velocity perpendicular to the magnetic field lines, as the field-aligned plasma flow is irrelevant[44]). Contributions from flux emergence and surface shearing motions are represented by the first and second terms, respectively. According to ref. 44, $\mathbf{V}_\perp = \mathbf{V} - (\mathbf{V} \cdot \mathbf{B})\mathbf{B}/B^2$, where $\mathbf{V}$ is the velocity vector derived by DAVE4VM. Similarly, the magnetic helicity flux across $S$ can be expressed by the combination of an emerging and a shearing terms[47]:

$$\left.\frac{dH}{dt}\right|_S = 2 \int_S (\mathbf{A}_p \cdot \mathbf{B}_t) V_{\perp n} dS - 2 \int_S (\mathbf{A}_p \cdot \mathbf{V}_{\perp t}) B_n dS, \tag{4}$$

where $\mathbf{A}_p$ is the vector potential of the potential field $\mathbf{B}_p$. As the helicity flux density is not a gauge invariant quantity, we study the helicity flux integrated over the whole AR. The Poynting and helicity fluxes derived with the DAVE4VM results based on SDO/HMI vector magnetograms have an uncertainty of 17% and 23%, respectively[44,45]. These were determined by ref. 44 using a Monte Carlo experiment where noises are randomly added to the HMI vector data. We also note that DAVE4VM has intrinsic method errors and may underestimate both Poynting and helicity fluxes by 29 and 10%, respectively[36,48].

**Software availability.** DAVE and DAVE4VM flow tracking codes as used in this study can be obtained from http://ccmc.gsfc.nasa.gov/lwsrepository/index.php.

**Data availability.** All the data used in the present study are publicly available. The BBSO/NST TiO and H-alpha images can be downloaded from http://bbso.njit.edu. The Fermi X-ray flux data can be downloaded from http://hesperia.gsfc.nasa.gov/fermi_solar. The GOES X-ray flux data can be downloaded from http://

www.ngdc.noaa.gov/stp/satellite/goes/dataaccess.html. The SDO/HMI vector magnetograms can be downloaded from http://jsoc.stanford.edu.

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

## Acknowledgements

We thank the BBSO, Fermi, GOES and SDO/HMI teams for providing the data. The BBSO operation is supported by NJIT, US NSF AGS 1250818 and NASA NNX13AG14G grants, and the NST operation is partly supported by the Korea Astronomy and Space Science Institute and Seoul National University, and by the strategic priority research programme of CAS with Grant Number XDB09000000. This work is supported by NASA under LWSTRT grants NNX13AF76G and NNX13AG13G, and HGI grants NNX14AC12G and NNX16AF72G, and by NSF under grants AGS 1250818, 1348513, 1408703 and 1539791. J.L. is supported by the BK21 Plus Program (21A20131111123) funded by the Ministry of Education (MOE, Korea) and National Research Foundation of Korea (NRF), and also by NRF-2012 R1A2A1A 03670387. This work uses the DAVE/DAVE4VM codes written and developed by P.W. Schuck at the Naval Research Laboratory.

## Author contributions

C.L. discovered the differential nature of this sunspot rotation, performed all the data analysis and interpretation, and wrote and revised the manuscript. Y.X. was the PI of this NST observation run and helped with the analysis of overall sunspot rotation. W.C. developed instruments of NST and coordinated the observation. N.D. contributed to the data processing and analysis, especially the flow tracking. J.L., H.S.H. and D.E.G. contributed to the result interpretation and presentation. J.W. and J.J. helped with the data processing. H.W. first observed this sunspot rotation and directed the research. All authors commented on the manuscript.

## Additional information

**Competing financial interests:** The authors declare no competing financial interests.

