## [Peer Review File · Nature Communications]

Reviewers' comments:

Reviewer #1 (Remarks to the Author):

A. Summary of Key Results

- High spatial and temporal resolution observations of a sunspot reveal irreversible changes following a large flare (M-class flare). The observational results provide strong evidence that changes in the solar corona (e.g. a flare) can have a significant impact on the solar photosphere.

B. Originality and Interest

- The main result in this paper can be considered an example of discovery science. The unique capabilities of the New Solar Telescope (NST) facilitated the discovery that sunspot structure in the photosphere changes in response to magnetic reconfigurations in the corona. For this reason the results presented should be considered novel and interesting to a diverse audience including solar physicists, magnetohydrodynamics (MHD) modelers and developers of astronomical instrumentation.

C. Data and Methodology

- The observations here were taken with the state-of-the-art NST at Big Bear Solar Observatory. With a primary mirror of 1.6 m, the NST is the largest solar telescope currently in operation. The large aperture size and the implementation of high-order adaptive optics allowed the authors to capture excellent quality observations with spatial resolutions of ~ 60 km on the solar surface and temporal cadence of 15 and 28 s (depending on the wavelength band). This high spatiotemporal resolution was essential for capturing the changes in sunspot morphology reported in the paper. The results presented are a preview of science results the Daniel K Inouye Solar Telescope (DKIST, with a 4m aperture) promises to deliver upon completion.

-The text and figures are well presented.

D. Appropriate use of statistics and treatment of uncertainties.

- Estimates of uncertainties of physical quantities derived from the observations are adequate.

E. Conclusions: Robustness, Validity, Reliability

- The report about irreversible sunspot umbra and penumbra changes following a flare is a robust.

- In line 194, the article states "Our observational results indicate that the flare energy release in the corona can cause bulk plasma motions on the photosphere, which is not deemed feasible by all contemporary flare models, and also does not favor some flare/CME models that do not require a coronal magnetic energy/storage."

I object to two points made in the previous statement. Firstly, the statement implies that pre-existing magnetic free energy in the corona is the energy source used to drive the observed photospheric rotation. This is not the only possibility. It may also be that the magnetic energy stored underneath the photosphere is the energy source for driving photospheric rotation. The way to distinguish between the two scenarios is to compute the vertical component of the Poynting flux.

A second objection to the statement is that ' [it] is not deemed feasible by all contemporary flare models'. In many flare models the photospheric boundary condition is forced to be line-tied and horizontal motion assumed zero. So by construction these models do address behavior reported in

this paper. However, this is different from claiming that the flare models deem such a phenomena unfeasible.

F. Suggested Improvements

- I suggest the authors consult the paper on magnetic field coupling between the corona and convection zone by Longcope & Welsch (2000). This paper considers how a vertical, twisted magnetic flux tube in the convection zone exists in an equilibrium state with the overlying coronal field, and how emergence of field into the corona creates torques that drive sunspot rotation. this type of idealized model may help elucidate the essential physics ideas needed to properly interpret the observations.

- I suggest the authors compute the Poynting flux during sunspot rotation to check whether the energy input is from the coronal or convection zone.

- The word "mass" in coronal mass ejection is missing in the abstract.

- In line 91 the SDO/AIA and SoHO/LASCO instruments should also be mentioned since the data is used in Extended Data Figure 2.

G. References

- Sufficient references are provided.

H. Clarify and context

- The article (including abstract, captions and main text) is lucid and concise and provides the appropriate level of context.

Reviewer #2 (Remarks to the Author):

This short paper shows unprecedented very interestingly analyzed data of sunspot rotation during and caused by a solar eruption. That alone is worth publishing since it revives the old discussions of the "tail that wags the dog" for coronal effects on the photosphere. This this paper presumably opens new questions in the field that ought to trigger a series of future papers aiming at interpreting this behavior.

Therefore I recommend this discovery to be published in Nature Communications.

Nevertheless, I recommend the authors first to revise and tone down in several places of there papers some of their interpretation-related sentences, as listed hereafter, that are either too restrictive, or ambiguous.

Once these issues are taken care of, the paper should, in my opinion, be published quickly.

- line 32 : Is $\beta = P/P_{\text{mag}} \gg 1$ inside sunspot umbrae ?
Theoretically speaking, in the framework of the approximate thin flux tube model, one can there reach $B=3500$ G at the depth of the Wilson depression if and only if the thermal pressure P tend to zero (see e.g. various reviews by Solanki). This question is not genuine, since its answer determines whether or not vector B changes in the photosphere can move the plasma through the Lorentz force (I come back to this point later in this report).

- line 39 : It is far not proven that CMEs are powered by magnetic reconnection, as claimed by the authors. Ideal loss of equilibrium and torus instability models for CMEs do not require reconnection to drive CMEs. CMEs are indeed associated with a flare-reconnection, but the latter does not necessarily cause the former.

- line 53 : In many confined/compact flares that result from loop-loop interactions in multipolar flux systems, there are always more than two ribbons, some of which moving towards PILs. The current sentence in the paper only refers to eruptive flares.

- line 58 : During their cooling phase, large loops are visible in EUV as well (as seen on almost every daily SDO movies), and often even in visible wavelengths such as H α when available.

- line 60 : Does "where the plasma dominates" imply $\beta \gg 1$?
If yes, then I ask the same question as above, for sunspot umbrae, that are considered in the present observations.

- lines 63-67 : "Only one recent theory" is just not true. The same behaviour as described in lines 65-66 is also produced by the reconnection-driven formation of flare loops in the standard flare model (see e.g. Janvier et al 2015 and Inoue et al 2015 and references therein). It has also been advocated by a flux rope collapse can produce the same effect (Petrie et al 2016 and references therein). Both theories produce an increase of horizontal fields down in the photosphere, each for different reasons. If β is not so large there, then flows may be accelerated through the Lorentz force. This ought be discussed as well.

- lines 132-151 : There I think the wording is too prudent/humble. The data neither "suggests" nor "seemingly" or "apparently" show what is described. To me the data unambiguously show all that, and that is a strength of the paper. The rest of the paper is "mere" analysis and numbering. But qualitatively speaking, the data is more than clear - at least to me.

- lines 195-197 : This sentence is a bit "out of the hat". It does not precisely address anything. Actually any MHD-based model of the corona can produce changes in the photospheric horizontal magnetic field without invoking photospheric flows, either vertical or horizontal. That's a natural consequence of the ideal-MHD induction equation. In addition, a few past publications have claimed that, to some degree, coronal velocities perturbations can be transmitted in the photosphere. So this part definitely needs to be rewritten.

- lines 232-236 : There is a big ambiguity (overstatement?) here regarding the ribbons and the CME. The former are flare-related. The latter is associated with loop expansion / dynamics that are not confined by the ribbons. Indeed, ribbons merely map the (quasi?) separatrix between the "opening" and the "reclosing" coronal loops. So one could easily argue that, since the sunspot rotation differentially follows the moving ribbon, it is related with the flare / reconnection, and not directly with the CME. So if the authors wish to maintain their "conjecture" about the CME-photosphere momentum conservation so-called "theory" (named as such in line 64), they should (1) develop it and explain why there is a link with flare ribbons [and it's not sufficient to write earlier that CMEs are powered by reconnection, since this is by far not proven] and (2) discuss not less-reasonable alternatives such as the reconnection-driven loop "reclosing" and the flux rope collapse, both of which being able to create flows in a not-so-high beta environment. In short, I believe that the authors oversell this momentum so called theory, and discard a bit too easily other well established (and maybe more natural?) reconnection-driven processes in eruptive flares.

- lines 237-240 : I am afraid I understand neither the proposed alternative interpretation, nor why it is discarded.

Reviewer #3 (Remarks to the Author):

>Who will be interested in reading the paper, and why?

The authors of the manuscript present and interpret observations of the rotating sunspot, where rotation is induced by an M6.5 solar flare, using unprecedented spatiotemporal resolution observations by New Solar Telescope (NST). These observations complement previously suggested theory and lower-resolution observations that solar flares could cause rotation of the sunspots where these flares occur. Since solar flares are major drivers of space weather, this paper will be interesting to all scientists interested in the mechanisms of solar flares and space weather in general.

> What are the main claims of the paper and how significant are they?

There are two claims presented in the paper: (1) observations of sudden sunspot rotation induced by a solar flare, using high-quality observations from the NST and (2) the details of the rotation profile, i.e. differential profile of rotation, and its relationship to flare ribbons. Both claims are important for understanding the flare mechanisms and are significant.

> Is the paper likely to be one of the five most significant papers published in the discipline this year?
Yes.

> How does the paper stand out from others in its field?

The results of the paper are extremely interesting and the observations presented are extraordinary.

> Are the claims novel? If not, which published papers compromise novelty?

The first claim, i.e. observations of flare-induced sunspot rotation, has been previously presented using lower-cadence observations from Solar Dynamics Observatory (Wang et al. 2014); however observations of NST describe the details of rotation profile in much more detail than has ever been done before. The second claim, i.e. the non-uniform rotation over the sunspot is novel.

> Are the claims convincing? If not, what further evidence is needed?

The first claim, i.e. sudden flare-induced sunspot rotation is convincing. There are minor discrepancies in the flare-rotation profiles, derived using different methods that need to be discussed. For more details please see below.

The second claim, i.e. differential rotation of sunspot and its relationship to ribbon location, needs more clear description in the text. Two panels (b,d) on Figure 1, used as a main proof for this claim are not very convincing, since ribbon shapes at two different times are quite similar and when plotted on different images are difficult to compare. To demonstrate how sunspot rotation is related to ribbons' location, it would be useful, for example, to show a dotted ribbon contour at two times on the same image or to have a figure (or a movie) showing ribbon locations overlaid onto vorticity maps at different times. For more details please see below.

> How much would further work improve it, and how difficult would this be? Would it take a long time?

All the proposed improvements are straightforward and should not take too long.

> Are the claims appropriately discussed in the context of previous literature?

Yes. The authors however should make it more clear that flare-induced sunspot rotation has been observed in the past (Wang et al. 2014) and that Lorentz force works (18,19) are widely accepted now by solar community and have found multiple confirmations in the observation.

General comments:

If we use angle evolution from the ellipse fitting (Top panel, Fig. 2) to derive angular speed of the ellipse, we will find that the maximum angular speed occurs some time around 18:10, significantly later than the peak time of the angular speed derived from the DAVE method (Middle Panel): $T \sim 17:58$. Please discuss this discrepancy.

Similarly there is a difference in the rotation profiles derived from ellipse fitting and polar remapping: rotation from ellipse fitting continues till Tend (Fig. 2), while rotation from polar remapping stops shortly after II (Fig. 3). Please discuss this discrepancy.

The structure of the paper could be improved. Some figures are shown long before they are described in the text (e.g. Panels b and d (Fig. 1); Panel c (Fig. 2)), affecting the clarity of the paper.

Specific Comments:

Line 46: Replace "currents driven" by "currents flowing"

Line 50: "It is generally accepted that accelerated particles can stream down from the coronal energy source along flaring magnetic loops to the low atmosphere"

Please replace with "It is generally accepted that accelerated particles can stream down from the reconnection site in the corona down to the low atmosphere along newly-formed magnetic loops".

Line 63. "Only one recent theory based on momentum conservation predicts that the photospheric magnetic field would become more horizontal (i.e., inclined to the surface) after flares/CMEs which has been seen in observations 20".

Please correct the text in the following way:

"Only one recent theory based on momentum conservation predicts that the photospheric magnetic field would become more horizontal (i.e., inclined to the surface) after flares/CMEs which has been seen in multiple observations (for example, 20)"

Line 75. Please correct M6.6 flare to M6.5 flare

Line 93. Please correct M6.6 flare to M6.5 flare

Line 93. Please correct "flare ribbon of which" to "flare ribbon which"

Line 95. Please refer to all figures in the same way: either Figures or Fig.

Figure 1.

Figure 1, Panels a-d. For the ease of reading please add titles to the top of each figure: for example, (a) BBSO/NST H-alpha, 22-Jun-2015 17:55:30 UT, (b) TiO Flow (DAVE) 22-Jun-2015 17:55:38, (c) BBSO/NST H-alpha, 22-Jun-2015 18:05:24 UT, (d) TiO Flow, 22-Jun-2015 18:05:24 (DAVE).

Figure 1, Panels b and d: 1) currently it is hard to see the arrows; try to make them thinner or shorter to see if it improves readability.

Figure 1, Panels b and d: 2) In the caption or in the text please describe which time difference has been used to derive the flow field.

Figure 1, Panels b and d: 3) From these two panels it is hard to see that the flows are associated with the ribbons, especially in f2. In the western part of f1 ribbons lie very close at two times, so that it is hard to track the change (dotted ribbon profiles at both times might help). A movie or a figure consisting of several panels showing ribbon boundaries overlaid on the vorticity map as a function of time would be useful to demonstrate this point.

Figure 1, Panels b and d: 4) These two panels are not discussed in the text until after Figure 2. It would be logical to either move discussion of Panels b & d to earlier location, or move the Figure forward.

Line 101: "closely associated with the flare. Such observation of a sudden sunspot rotation"
I would recommend highlighting here that the rotation speeds in locations swept by flare ribbons, i.e. above the white line are larger, than the ones below it, implying differential rotation of the sunspot. Otherwise it is not clear further in text (line 108) what is meant by observed differential rotation.

Line 125: Please replace "that can be modeled" and "the model indicates" with "that can be approximated" and "this fit indicates" accordingly.

Line 127: Please replace "through a total of 9" by "by a total of 9"

Line 152. See comment (4) to Figure 1, Panels b and d.

Line 168: "(see Supplementary Video 5) indicating that the sunspot rotation is intimately linked to the flaring process"

1) Please rename videos so that they have descriptive names. Please refer to videos using meaningful name, e.g. instead of "(see Supplementary Video 5)" use "(see Supplementary Video 5, Vorticity evolution)"

2) Video 5 only shows vorticity evolution. From looking at the video it is not clear how vorticity is related to ribbons' location. Please add an overlay of the ribbons location to the video.

Line 175: "Along with the flare ribbon evolution (Supplementary Video 3), it is obvious that ... features 1-4 and ... 5-7 ... start rotating as the ribbon sweeps"

From Figure 3 it is not clear how features 1-4 and 5-7 are located relative to ribbons, since ribbons are not shown in Figure 3. Asterisks could be added to Figure 3 to show ribbons' locations at certain times to prove this point.

Line 181: "angular speed up to ~50 degrees/hour, much higher"

Please correct to "angular speed up to ~50 degrees/hour (e.g. dotted line of feature 3), much higher"

Figure 3: Please add angular speeds to the dotted lines next to feature numbers. Otherwise it might be not evident to the reader where the estimate of 50 deg/hr comes from.

Figure 3: Figure 3 suggests that rotation stops around 18:10. However Top panel of Figure 2 shows that rotation continues till Tend. Please explain.

Line 210: "Indeed, for spot f1"

It is unclear what "indeed" here refers to, since intensity does not seem to be related to the discussion above.

Line 426: "The error is represented by 1 s.d. ...evolution."

This is not very clear. Please explain how the errors are calculated here, since they seem much smaller on the plot than in the movie.

Line 455: "The error is represented by 1 s.d."

Please see the previous comment.

Extended Data Figure 1: Please add titles to each panel.

Extended Data Figure 2: Please add titles to each panel.

Extended Data Figure 2: Please decrease thickness of the arrows.

Extended Data Figure 2: Adding ribbon contour could be useful to describe the relationship between Lorentz forces and ribbons.

Extended Data Figure 2: Two bottom panels don't seem to be essential to the main point of the paper and could be omitted.

Videos: Please rename videos so that they have descriptive names.

Video 5: Please add color scale with units.

Point-to-point response to referees' comments (NCOMMS-16-07180-T)

We appreciate the referees' comments that help us to improve the paper. We give reply to each of the comments as follows. New line numbers corresponding to referees' comments are provided in replies.

In the revised paper, we restructured the whole manuscript for a better presentation, strengthened the content to include more physical discussions, and modified the format so that it conforms to the requirements of Nature Communications. We used blue color to highlight the major changes, most of which are after referees' comments; for a section with many changes, the section title is colored blue.

Reviewer #1 (Remarks to the Author):

A. Summary of Key Results

- High spatial and temporal resolution observations of a sunspot reveal irreversible changes following a large flare (M-class flare). The observational results provide strong evidence that changes in the solar corona (e.g. a flare) can have a significant impact on the solar photosphere.

B. Originality and Interest

- The main result in this paper can be considered an example of discovery science. The unique capabilities of the New Solar Telescope (NST) facilitated the discovery that sunspot structure in the photosphere changes in response to magnetic reconfigurations in the corona. For this reason the results presented should be considered novel and interesting to a diverse audience including solar physicists, magnetohydrodynamics (MHD) modelers and developers of astronomical instrumentation.

C. Data and Methodology

- The observations here were taken with the state-of-the-art NST at Big Bear Solar Observatory. With a primary mirror of 1.6 m, the NST is the largest solar telescope currently in operation. The large aperture size and the implementation of high-order adaptive optics allowed the authors to capture excellent quality observations with spatial resolutions of ~ 60 km on the solar surface and temporal cadence of 15 and 28 s (depending on the wavelength band). This high spatiotemporal resolution was essential for capturing the changes in sunspot morphology reported in the paper. The results presented are a preview of science results the Daniel K Inouye Solar Telescope (DKIST, with a 4m aperture) promises to deliver upon completion.

-The text and figures are well presented.

D. Appropriate use of statistics and treatment of uncertainties.

- Estimates of uncertainties of physical quantities derived from the observations are adequate.

E. Conclusions: Robustness, Validity, Reliability

- The report about irreversible sunspot umbra and penumbra changes following a flare is a robust.

- In line 194, the article states "Our observational results indicate that the flare energy release in the corona can cause bulk plasma motions on the photosphere, which is not deemed feasible by all contemporary flare models, and also does not favor some flare/CME models that do not require a coronal magnetic energy/storage."

I object to two points made in the previous statement. Firstly, the statement implies that pre-existing magnetic free energy in the corona is the energy source used to drive the observed photospheric

rotation. This is not the only possibility. It may also be that the magnetic energy stored underneath the photosphere is the energy source for driving photospheric rotation. The way to distinguish between the two scenarios is to compute the vertical component of the Poynting flux.

REPLY: In lines 299-325 and the Discussion section, we added the computation of the vertical component of the Poynting flux and related discussions. The result indicates that the energy source driving the sunspot rotation comes from the corona.

A second objection to the statement is that '[it] is not deemed feasible by all contemporary flare models'. In many flare models the photospheric boundary condition is forced to be line-tied and horizontal motion assumed zero. So by construction these models do address behavior reported in this paper. However, this is different from claiming that the flare models deem such a phenomena unfeasible.

REPLY: We removed this sentence.

F. Suggested Improvements

- I suggest the authors consult the paper on magnetic field coupling between the corona and convection zone by Longcope & Welsch (2000). This paper considers how a vertical, twisted magnetic flux tube in the convection zone exists in an equilibrium state with the overlying coronal field, and how emergence of field into the corona creates torques that drive sunspot rotation. this type of idealized model may help elucidate the essential physics ideas needed to properly interpret the observations.

REPLY: In lines 327-360, we added the related discussions based on Longcope & Welsch (2000) and Fan (2009).

- I suggest the authors compute the Poynting flux during sunspot rotation to check whether the energy input is from the coronal or convection zone.

REPLY: In lines 299-325, we computed the Poynting flux and found that the energy input is from the corona.

- The word "mass" in coronal mass ejection is missing in the abstract.

REPLY: The term "coronal mass ejection" is now not mentioned in the Abstract.

- In line 91 the SDO/AIA and SoHO/LASCO instruments should also be mentioned since the data is used in Extended Data Figure 2.

REPLY: For a more focused discussion, we removed the materials based on AIA and LASCO images.

G. References

- Sufficient references are provided.

H. Clarify and context

- The article (including abstract, captions and main text) is lucid and concise and provides the appropriate level of context.

Reviewer #2 (Remarks to the Author):

This short paper shows unprecedented very interestingly analyzed data of sunspot rotation during and caused by a solar eruption. That alone is worth publishing since it revives the old discussions of the "tail that wags the dog" for coronal effects on the photosphere. This this paper presumably opens new questions in the field that ought to trigger a series of future papers aiming at interpreting this behavior.

Therefore I recommend this discovery to be published in Nature Communications.

Nevertheless, I recommend the authors first to revise and tone down in several places of there papers some of their interpretation-related sentences, as listed hereafter, that are either too restrictive, or ambiguous.

Once these issues are taken care of, the paper should, in my opinion, be published quickly.

- line 32 : Is $\beta = P/P_{\text{mag}} \gg 1$ inside sunspot umbrae ?
Theoretically speaking, in the framework of the approximate thin flux tube model, one can there reach $B=3500$ G at the depth of the Wilson depression if and only if the thermal pressure P tend to zero (see e.g. various reviews by Solanki). This question is not genuine, since its answer determines wether or not vector B changes in the photosphere can move the plasma through the Lorentz force (I come back to this point later in this report).

REPLY: In lines 1-8: At the continuum photospheric level, the plasma-beta evolves from lower than unity in the umbra to larger than unity in the outer penumbra; overall, $\beta \sim 1$ within sunspots (Borrero & Ichimoto 2011, Sect. 2.4, and references therein). We added the references and amended the description.

- line 39 : It is y far not proven that CMEs are powered by magnetic reconnection, as claimed by the authors. Ideal loss of equilibrium and torus instability models for CMEs do not require reconnection to drive CMEs. CMEs are indeed associated with a flare-reconnection, but the latter does not necessarily cause the former.

REPLY: In line 12, we removed the mentioning of CMEs from this sentence.

- line 53 : In many confined/compact flares that result from loop-loop interactions in multipolar flux systems, there are always more than two ribbons, some of which moving towards PILs. The current sentence in the paper only refers to eruptive

flares.

REPLY: In lines 24-25, we constrained our description to eruptive flares.

- line 58 : During their cooling phase, large loops are visible in EUV as well (as seen on almost every daily SDO movies), and often even in visible wavelengths such as H α when available.

REPLY: In line 30, we revised the description to include both SXR and other emissions.

- line 60 : Does "where the plasma dominates" imply $\beta \gg 1$? If yes, then I ask the same question as above, for sunspot umbrae, that are considered in the present observations.

REPLY: In line 32, we removed "where the plasma dominates".

- lines 63-67 : "Only one recent theory" is just not true. The same behaviour as described in lines 65-66 is also produced by the reconnection-driven formation of flare loops in the standard flare model (see e.g. Janvier et al 2015 and Inoue et al 2015 and references therein). It has also been advocated by a flux rope collapse can produce the same effect (Petrie et al 2016 and references therein). Both theories produce an increase of horizontal fields down in the photosphere, each for different reasons. If β is not so large there, then flows may be accelerated through the Lorentz force. This ought be discussed as well.

REPLY: We agree that this behavior (i.e., photospheric field becomes more horizontal after flares) can be accommodated by the standard flare models as a result of newly formed flare loops (as discussed in Janvier et al. 2015 and Inoue et al. 2015). This was also pointed out in our previous studies (e.g., Wang & Liu 2010, ApJ, 716, L195; Liu et al. 2012, ApJ, 745, L4). Nevertheless, we note that the standard model was not established for explaining this behavior explicitly; in contrast, the model proposed by Hudson et al. (2008) and further developed by Fisher et al. (2012) could perhaps be the only one that explicitly predicts this behavior from the theoretical treatment. The "flux rope collapse" scenario as discussed in Petrie et al. (2016) is also based on the theory of Hudson et al. and Fisher et al.

In lines 36-37, for a simplified description, in this revision we modified this sentence to begin with "One particular recent theory...".

In lines 42-46, we also added the description that as the plasma-beta is smaller than unity in umbrae and inner penumbrae, flows may be accelerated through the Lorentz force change, which is quantified in the theory of Hudson et al. and Fisher et al.

- lines 132-151 : There I think the wording is too prudent/humble. The data neither "suggests" nor "seemingly" or "apparently" show what is described. To me the data unambiguously show all that, and that is a strength of the paper. The rest of the paper is "mere" analysis and numbering. But qualitatively speaking, the data is more than clear - at least to me.

REPLY: In lines 143-144, we changed the description to read as follows: "...full-resolution movies ...unambiguously show that..."

- lines 195-197 : This sentence is a bit "out of the hat". It does not precisely address anything. Actually any MHD-based model of the corona can produce changes in the photospheric horizontal magnetic field without invoking photospheric flows, either vertical or horizontal. That's a natural consequence of the ideal-MHD induction equation. In addition, a few past publications have claimed that, to some degree, coronal velocities perturbations can be transmitted in the photosphere. So this part definitely needs to be rewritten.

REPLY: We removed this sentence.

- lines 232-236 : There is a big ambiguity (overstatement?) here regarding the ribbons and the CME. The former are flare-related. The latter is associated with loop expansion / dynamics that are not confined by the ribbons. Indeed, ribbons merely map the (quasi?) separatrix between the "opening" and the "reclosing" coronal loops. So one could easily argue that, since the sunspot rotation differentially follows the moving ribbon, it is related with the flare / reconnection, and not directly with the CME. So if the authors wish to maintain their "conjecture" about the CME-photosphere momentum conservation so-called "theory" (named as such in line 64), they should (1) develop it and explain why there is a link with flare ribbons [and it's not sufficient to write earlier that CMEs are powered by reconnection, since this is by far not proven] and (2) discuss not less-reasonable alternatives such as the reconnection-driven loop "reclosing" and the flux rope collapse, both of which being able to create flows in a not-so-high beta environment. In short, I believe that the authors oversell this momentum so called theory, and discard a bit too easily other well established (and maybe more natural?) reconnection-driven processes in eruptive flares.

REPLY: We removed these discussions related to CME-photosphere momentum conservation. Again we note that the reconnection-driven loop "reclosing" and the flux rope collapse scenarios are both in line with the theory of Hudson et al. (2008) and Fisher et al. (2012); please see the above response to "- lines 63-67"

- lines 237-240 : I am afraid I understand neither the proposed alternative interpretation, nor why it is discarded.

REPLY: In lines 327-360, we have provided related discussions based on the work of Longcope & Welsch (2000) and Fan (2009).

Reviewer #3 (Remarks to the Author):

> Who will be interested in reading the paper, and why?

The authors of the manuscript present and interpret observations of the rotating sunspot, where rotation is induced by an M6.5 solar flare, using unprecedented spatiotemporal resolution observations by New Solar Telescope (NST). These observations complement previously suggested theory and lower-resolution observations that solar flares could cause rotation of the sunspots where these flares occur. Since solar flares are major drivers of space weather, this paper will be interesting to all scientists interested in the mechanisms of solar flares and space weather in general.

> What are the main claims of the paper and how significant are they?

There are two claims presented in the paper: (1) observations of sudden sunspot rotation induced by a solar flare, using high-quality observations from the NST and (2) the details of the rotation profile, i.e. differential profile of rotation, and its relationship to flare ribbons. Both claims are important for understanding the flare mechanisms and are significant.

> Is the paper likely to be one of the five most significant papers published in the discipline this year?
Yes.

> How does the paper stand out from others in its field?

The results of the paper are extremely interesting and the observations presented are extraordinary.

> Are the claims novel? If not, which published papers compromise novelty?

The first claim, i.e. observations of flare-induced sunspot rotation, has been previously presented using lower-cadence observations from Solar Dynamics Observatory (Wang et al. 2014); however observations of NST describe the details of rotation profile in much more detail than has ever been done before. The second claim, i.e. the non-uniform rotation over the sunspot is novel.

> Are the claims convincing? If not, what further evidence is needed?

The first claim, i.e. sudden flare-induced sunspot rotation is convincing. There are minor discrepancies in the flare-rotation profiles, derived using different methods that need to be discussed. For more details please see below.

The second claim, i.e. differential rotation of sunspot and its relationship to ribbon location, needs more clear description in the text. Two panels (b,d) on Figure 1, used as a main proof for this claim are not very convincing, since ribbon shapes at two different times are quite similar and when plotted on different images are difficult to compare. To demonstrate how sunspot rotation is related to ribbons' location, it would be useful, for example, to show a dotted ribbon contour at two times on the same image or to have a figure (or a movie) showing ribbon locations overlaid onto vorticity maps at different times. For more details please see below.

REPLY: We added a new Fig. 4 in which ribbons locations are overlaid on vorticity maps at different times. We note that different sections of the ribbon did not propagate at the same speed. In fact, the northern portion of the ribbon moves much faster than the southern part, as can be seen in Fig.4. We mentioned this in line 184.

> How much would further work improve it, and how difficult would this be? Would it take a long time?

All the proposed improvements are straightforward and should not take too long.

> Are the claims appropriately discussed in the context of previous literature?

Yes. The authors however should make it more clear that flare-induced sunspot rotation has been observed in the past (Wang et al. 2014) and that Lorentz force works (18,19) are widely accepted now by solar community and have found multiple confirmations in the observation.

REPLY: In lines 41-42 and 47-49, we have improved the related descriptions.

General comments:

If we use angle evolution from the ellipse fitting (Top panel, Fig. 2) to derive angular speed of the ellipse, we will find that the maximum angular speed occurs some time around 18:10, significantly later than the peak time of the angular speed derived from the DAVE method (Middle Panel): $T \sim 17:58$. Please discuss this discrepancy.

REPLY: First, we have improved the ellipse fitting (please see lines 405-409). Previously, we used the IDL function MPFITELLIPSE, which fits an ellipse to the contour of sunspot boundary. In this revision, we use another function FIT_ELLIPSE, which does a “center of mass” ellipse fitting based on indices of the sunspot region itself. The improved result is shown in Fig.2. Now the time profile of the orientation angle can be best approximated using an acceleration phase during 17:56 and 18:12:29 UT and a deceleration phase afterward.

Second, the ellipse fitting is based on the assumption of a solid-body rotation, while the DAVE results study the flows within the sunspot umbra to reveal the local differential rotation. The DAVE method gives a peak time of angular speed around 17:56:23 UT; however, only the western portion of the umbra f1 rotates at this stage (see discussions in lines 163-179). Since internal rotation of a portion of the sunspot does not necessarily imply the rotation of the sunspot as a whole (solid-body approximation), the results from these two approaches may not be compared directly.

In this revision, we discussed in lines 179-182 that “It is pertinent to point out that the afore-described ellipse fitting under a solid-body assumption shows a significant rotation of f1 only after $\sim 17:56$ UT. This highlights the differential nature of this sunspot rotation.”

Similarly there is a difference in the rotation profiles derived from ellipse fitting and polar remapping: rotation from ellipse fitting continues till T_{end} (Fig. 2), while rotation from polar remapping stops shortly after II (Fig. 3). Please discuss this discrepancy.

REPLY: The polar remapping can only detect local rotations, not the rotation of the entire sunspot as a solid body (this is studied by the ellipse fitting). In this revision, we provided discussions in lines 203-209 that during phase-3 (after the HXR peak III), “The rotational flows involving both f1 and f2 diminish, as reflected by the observations that the mean vorticity of f1/f2 largely returns to the pre-flare level (Fig. 5) and that drifting features nearly flattens in the re-mapped space-time slice images (Fig. 6). Interestingly, f1 shows overall westward and southwestward flows (Fig.3f), and it continues to rotate clockwise as a whole (Fig.2).

The structure of the paper could be improved. Some figures are shown long before they are described in the text (e.g. Panels b and d (Fig. 1); Panel c (Fig. 2)), affecting the clarity of the paper.

REPLY: We have restructured the whole paper, so that figures and related text are presented in almost the same pace.

Specific Comments:

Line 46: Replace "currents driven" by "currents flowing"

REPLY: [line 18] Replaced.

Line 50: "It is generally accepted that accelerated particles can stream down from the coronal energy source along flaring magnetic loops to the low atmosphere"

Please replace with "It is generally accepted that accelerated particles can stream down from the reconnection site in the corona down to the low atmosphere along newly-formed magnetic loops".

REPLY: [lines 21-23] Replaced.

Line 63. "Only one recent theory based on momentum conservation predicts that the photospheric magnetic field would become more horizontal (i.e., inclined to the surface) after flares/CMEs which has been seen in observations 20".

Please correct the text in the following way:

"Only one recent theory based on momentum conservation predicts that the photospheric magnetic field would become more horizontal (i.e., inclined to the surface) after flares/CMEs which has been seen in multiple observations (for example, 20)"

REPLY: In lines 36-42, we have improved the descriptions.

Line 75. Please correct M6.6 flare to M6.5 flare

REPLY: [line 53] Corrected.

Line 93. Please correct M6.6 flare to M6.5 flare

REPLY: [line 94] Corrected.

Line 93. Please correct "flare ribbon of which" to "flare ribbon which"

REPLY: In lines 100-106, we improved the related description.

Line 95. Please refer to all figures in the same way: either Figures or Fig.

REPLY: We checked and followed the General Figure Guidelines of Nature Communications to abbreviate "Figure" as "Fig." and "Figures" as "Figs" except at the start of a sentence, and refer to figures after the main text as "Supplementary Fig.".

(**) The guideline also requires that the associated movies are referred in the main text as "Supplementary Movie", and that each movie is accompanied by the legend (including a title and description). The legends of all Supplementary Movies are provided at the end of the manuscript.

Figure 1.

Figure 1, Panels a-d. For the ease of reading please add titles to the top of each figure: for example, (a) BBSO/NST H-alpha, 22-Jun-2015 17:55:30 UT, (b) TiO Flow (DAVE) 22-Jun-2015 17:55:38, (c) BBSO/NST H-alpha, 22-Jun-2015 18:05:24 UT, (d) TiO Flow, 22-Jun-2015 18:05:24 (DAVE).

REPLY: Done for the new Fig. 3.

Figure 1, Panels b and d: 1) currently it is hard to see the arrows; try to make them thinner or shorter to see if it improves readability.

REPLY: Done for the new Fig. 3.

Figure 1, Panels b and d: 2) In the caption or in the text please describe which time difference has been used to derive the flow field.

REPLY: Done for the new Fig. 3.

Figure 1, Panels b and d: 3) From these two panels it is hard to see that the flows are associated with the ribbons, especially in f2. In the western part of f1 ribbons lie very close at two times, so that it is hard to track the change (dotted ribbon profiles at both times might help). A movie or a figure consisting of several panels showing ribbon boundaries overlaid on the vorticity map as a function of time would be useful to demonstrate this point.

REPLY: First, the southern (western) part of the ribbon across f1 lies very close at multiple times because this portion of the ribbon moves much slower than the northern (eastern) part of the ribbon. Second, we provide a new Fig. 4, in which vorticity maps are overlaid with ribbon fronts. Related discussions are mainly given in lines 222-229.

Figure 1, Panels b and d: 4) These two panels are not discussed in the text until after Figure 2. It would be logical to either move discussion of Panels b & d to earlier location, or move the Figure forward.

REPLY: We restructured the paper so that all figures are presented in almost the same pace as their related discussions.

Line 101: "closely associated with the flare. Such observation of a sudden sunspot rotation"
I would recommend highlighting here that the rotation speeds in locations swept by flare ribbons, i.e. above the white line are larger, than the ones below it, implying differential rotation of the sunspot. Otherwise it is not clear further in text (line 108) what is meant by observed differential rotation.

REPLY: In line 146, we now clearly state before detailed discussions that "...as the flare ribbon moves across, different portions of the sunspot start rotating at different times (*meaning a differential rotation*) corresponding to the peaks of HXR emission".

Line 125: Please replace "that can be modeled" and "the model indicates" with "that can be approximated" and "this fit indicates" accordingly.

REPLY: [line 139] Replaced.

Line 127: Please replace "through a total of 9" by "by a total of 9"

REPLY: [line 135] Revised.

Line 152. See comment (4) to Figure 1, Panels b and d.

REPLY: We provide a new Fig. 4, in which vorticity maps are overlaid with ribbon fronts. Related discussions are mainly given in lines 222-229.

Line 168: "(see Supplementary Video 5) indicating that the sunspot rotation is intimately linked to the flaring process"

1) Please rename videos so that they have descriptive names. Please refer to videos using meaningful name, e.g. instead of "(see Supplementary Video 5)" use "(see Supplementary Video 5, Vorticity evolution)"

REPLY: [line 224] Done. Note we now use "Supplementary Movie" instead of "Supplementary Video" to conform to this journal's guide line. Also please see the above reply marked by (**).

2) Video 5 only shows vorticity evolution. From looking at the video it is not clear how vorticity is related to ribbons' location. Please add an overlay of the ribbons location to the video.

REPLY: Following the previous comments, we provide a new Fig. 4, in which we overlay ribbon fronts on vorticity maps at selected times. The flare ribbon fronts in these high-resolution images may not be easily extracted using a simple contouring method, thus overlay of the ribbon front onto every frame of the video is difficult.

Line 175: "Along with the flare ribbon evolution (Supplementary Video 3), it is obvious that ... features 1-4 and ... 5-7 ... start rotating as the ribbon sweeps"

From Figure 3 it is not clear how features 1-4 and 5-7 are located relative to ribbons, since ribbons are not shown in Figure 3. Asterisks could be added to Figure 3 to show ribbons' locations at certain times to prove this point.

REPLY: In the new Fig.4, we present features 1-4 and 5-7 together with the ribbon front. This facilitates the related discussions, such as in lines 170-172.

Line 181: "angular speed up to ~50 degrees/hour, much higher"

Please correct to "angular speed up to ~50 degrees/hour (e.g. dotted line of feature 3), much higher"

REPLY: In lines 173 and 190, we specify that these are mean angular velocities of features. In the caption of Fig. 6, we describe that "The black dotted lines trace several distinct features 1-10 in fl by a linear approximation. The numbers in bracket are the corresponding angular velocity (in degree per hour) from the linear fit".

Figure 3: Please add angular speeds to the dotted lines next to feature numbers. Otherwise it might be not evident to the reader where the estimate of 50 deg/hr comes from.

REPLY: Done in the new Fig. 6.

Figure 3: Figure 3 suggests that rotation stops around 18:10. However Top panel of Figure 2 shows that rotation continues till Tend. Please explain.

REPLY: We provided discussions in line 203-209 that during phase-3 (after the HXR peak III), "The rotational flows involving both f1 and f2 diminish, as reflected by the observations that the mean vorticity of f1/f2 largely returns to the pre-flare level (Fig. 5) and that drifting features nearly flattens in the re-mapped space-time slice images (Fig. 6). Interestingly, f1 shows overall westward and

southwestward flows (Fig.3f), and it continues to rotate clockwise as a whole (Fig.2). Again, as we explained previously for earlier comments, the polar-remapping detects local rotation while the ellipse-fitting detects the larger scale rotation under a solid-body approximation.

Line 210: "Indeed, for spot fl"

It is unclear what "indeed" here refers to, since intensity does not seem to be related to the discussion above.

REPLY: We removed the word "Indeed" and improved the related discussions in lines 264-266.

Line 426: "The error is represented by 1 s.d. ...evolution."

This is not very clear. Please explain how the errors are calculated here, since they seem much smaller on the plot than in the movie.

REPLY: We now follow the journal guide line to provide descriptions of errors in figure captions. Thus in the caption of Fig. 5, we state that "The error bars (plotted every 1 minute to better show the results) represent 1 s.d. from the average mean-vorticity over a pre-flare period (17:00 to 17:39 UT), demonstrating the significant flare-related variations compared to the variation seen in the long-term evolution."

Line 455: "The error is represented by 1 s.d."

Please see the previous comment.

REPLY: In the caption of Fig. 8, we state that "The error bars represent 1 s.d. calculated from the provided uncertainty of HMI vector field."

Extended Data Figure 1: Please add titles to each panel.

REPLY: This figure now becomes Fig. 1 in the main text. Titles are added.

Extended Data Figure 2: Please add titles to each panel.

REPLY: Part of this figure now becomes Fig. 7 in the main text. Titles are added.

Extended Data Figure 2: Please decrease thickness of the arrows.

REPLY: Done for the new Fig. 7.

Extended Data Figure 2: Adding ribbon contour could be useful to describe the relationship between Lorentz forces and ribbons.

REPLY: Done for the new Fig. 7. Related discussions are added in lines 282-285.

Extended Data Figure 2: Two bottom panels don't seem to be essential to the main point of the paper and could be omitted.

REPLY: Omitted.

Videos: Please rename videos so that they have descriptive names.

REPLY: Done.

Video 5: Please add color scale with units.

REPLY: Done.

Other Main Changes:

- We followed the format guideline of this journal to (1) revise the first paragraph and make it the Abstract of this paper, which has a 150 word limit and should contain no reference, (2) include in the last paragraph of Introduction “a brief summary of both the results and the conclusions (written in present tense)”, and (3) divide the Results and Methods sections by topical subheadings.

- Following the guideline for figures, we (1) included a description of the error bars in the figure legend when applicable, and (2) used a clear, sans-serif typeface (Helvetica) for text in figures. In addition, we changed the figure axis unit from Mm to arcsec, to be consistent with most solar physics papers published in this and other journals.

- Lines 506-517: We added the required Data Availability and Software Availability statements at the end of Methods.

REVIEWERS' COMMENTS:

Reviewer #1 (Remarks to the Author):

The authors have addressed my concerns in the original report. I recommend the paper for acceptance.

Reviewer #2 (Remarks to the Author):

All the context issues that I had mentioned have now be fine-tuned in the text. This, along with the answers to the other referee's comments, make this discovery paper extremely interesting. Therefore I am now glad to recommend its publication in Nature Communications.

Reviewer #3 (Remarks to the Author):

I would like to thank the authors for a very careful revision of the manuscript where all referee's comments have been addressed.

Below are several minor comments that appeared in response to the new material in the manuscript.

>37 One particular recent theory based on momentum conservation
>38 predicts that as a back reaction on the solar surface and in-
>39 terior, the photospheric magnetic field would become more
>40 horizontal (i.e., inclined to the surface) near flaring PILs after
>flares/CMEs^{16, 17, 41}, which has been seen in multiple observations
>(for example, refs^{18–21, 42}).

To improve the clarity of this sentence I would recommend to reorganize it.

>355 Under this scenario, it would be expected that the Poynting flux E'
>356 and also helicity flux H'
>357 (with the same sign as that before the eruption) injected
>358 into the atmosphere by the emerging flux tube would also
>359 suddenly enhance.

Please add reference to Kazachenko et al. 2015, currently the most detailed study of Poynting fluxes in evolving active region. In Kazachenko et al. 2015 the authors found a flare-associated increase in the total Poynting flux both spatially (Fig. 8, lower panel) and when integrated over the field of view (Fig. 9, Panel C).

>501 The Poynting and helicity fluxes derived with the DAVE4VM
>502 results based on SDO/HMI vector magnetograms have an
>503 uncertainty of 17% and 23%, respectively^{42, 43}. These were
>503 determined by ref. 42 using a Monte Carlo experiment where
>504 noises are randomly added to the HMI vector data.

Note that these error estimates account for uncertainties in the input data only, and not the method itself. Using an ANMHD test case, where both energy and helicity fluxes are known, it has been shown, that, as any other method (Welsch et al. 2007), DAVE4VM has intrinsic method errors (Schuck 2008, Kazachenko et al., 2014). According to Table 4 in Kazachenko et al. 2014, DAVE4VM underestimated both Poynting and helicity fluxes by 29% (fraction=0.71) and 10% (fraction=0.9) respectively.

Point-to-point response to referees' comments (NCOMMS-16-07180-A)

We appreciate the referees' further comments that help us to improve the accuracy of the paper. We give reply to each of the comments as follows. We used blue color to highlight the changes in the text.

REVIEWERS' COMMENTS:

Reviewer #1 (Remarks to the Author):

The authors have addressed my concerns in the original report. I recommend the paper for acceptance.

Reviewer #2 (Remarks to the Author):

All the context issues that I had mentioned have now be fine-tuned in the text. This, along with the answers to the other referee's comments, make this discovery paper extremely interesting. Therefore I am now glad to recommend its publication in Nature Communications.

Reviewer #3 (Remarks to the Author):

I would like to thank the authors for a very careful revision of the manuscript where all referee's comments have been addressed.

Below are several minor comments that appeared in response to the new material in the manuscript.

>37 One particular recent theory based on momentum conservation
>38 predicts that as a back reaction on the solar surface and in-
>39 terior, the photospheric magnetic field would become more
>40 horizontal (i.e., inclined to the surface) near flaring PILs after
>flares/CMEs16, 17 41 , which has been seen in multiple observations
>(for example, refs18–21 42).

To improve the clarity of this sentence I would recommend to reorganize it.

REPLY: Reorganized. The long sentence has been split into two, and the clarity has been improved.

>355 Under this scenario, it would be expected that the Poynting flux E'
>356 and also helicity flux H'
>357 (with the same sign as that before the eruption) injected
>358 into the atmosphere by the emerging flux tube would also
>359 suddenly enhance.

Please add reference to Kazachenko et al. 2015, currently the most detailed study of Poynting fluxes in evolving active region. In Kazachenko et al. 2015 the authors found a flare-associated increase in the total Poynting flux both spatially (Fig. 8, lower panel) and when integrated over the field of view (Fig. 9, Panel C).

REPLY: We have added the reference Kazachenko et al. 2015 to this sentence.

>501 The Poynting and helicity fluxes derived with the DAVE4VM
>502 results based on SDO/HMI vector magnetograms have an

>503 uncertainty of 17% and 23%, respectively 42, 43. These were
>503 determined by ref. 42 using a Monte Carlo experiment where
>504 noises are randomly added to the HMI vector data.

Note that these error estimates account for uncertainties in the input data only, and not the method itself. Using an ANMHD test case, where both energy and helicity fluxes are known, it has been shown, that, as any other method (Welsch et al. 2007), DAVE4VM has intrinsic method errors (Schuck 2008, Kazachenko et al., 2014). According to Table 4 in Kazachenko et al. 2014, DAVE4VM underestimated both Poynting and helicity fluxes by 29% (fraction=0.71) and 10% (fraction=0.9) respectively.

REPLY: We have added the following sentence to follow the above description. “We also note that DAVE4VM has intrinsic method errors and may underestimate both Poynting and helicity fluxes by 29% and 10%, respectively (Schuck08; Kazachenko et al. 2014)”.